# Mining and Tailings Dam Detection in Satellite Imagery Using Deep Learning

**DOI:** 10.3390/s20236936

**Published:** 2020-12-04

**Authors:** Remis Balaniuk, Olga Isupova, Steven Reece

**Affiliations:** 1Graduate Program in Governance, Technology and Innovation, Universidade Católica de Brasília, Brasília 71966-700, Brazil; 2Department Computer Science, University of Bath, Bath BA2 7PB, UK; oi260@bath.ac.uk; 3Department Engineering Science, Oxford University, Oxford OX1 3PJ, UK; reece@robots.ox.ac.uk

**Keywords:** tailings dam detection, surface mines detection, environmental impact of mining, remote sensing, machine learning, deep learning, cloud computing

## Abstract

This work explores the combination of free cloud computing, free open-source software, and deep learning methods to analyze a real, large-scale problem: the automatic country-wide identification and classification of surface mines and mining tailings dams in Brazil. Locations of officially registered mines and dams were obtained from the Brazilian government open data resource. Multispectral Sentinel-2 satellite imagery, obtained and processed at the Google Earth Engine platform, was used to train and test deep neural networks using the TensorFlow 2 application programming interface (API) and Google Colaboratory (Colab) platform. Fully convolutional neural networks were used in an innovative way to search for unregistered ore mines and tailing dams in large areas of the Brazilian territory. The efficacy of the approach is demonstrated by the discovery of 263 mines that do not have an official mining concession. This exploratory work highlights the potential of a set of new technologies, freely available, for the construction of low cost data science tools that have high social impact. At the same time, it discusses and seeks to suggest practical solutions for the complex and serious problem of illegal mining and the proliferation of tailings dams, which pose high risks to the population and the environment, especially in developing countries.

## 1. Introduction

About 3.5 billion people live in countries rich in oil, gas, or minerals (https://www.worldbank.org/en/topic/extractiveindustries). Natural resources have the potential to drive growth, progress, and poverty reduction in developing countries. Despite its economic importance, mining is associated with major risks, both related to accidents and environmental damage. Tailings dams are a by-product of mining, storing toxic waste liquid and solids. The failure of tailings dams can have a catastrophic effect, with huge amounts of water and solid material being released suddenly, which can cause great loss of human life and huge damage to the environment, buildings, and plantations. The risks posed by long-term containment and the high number of recent tailings dam failures have led to a growing awareness of the need for improved safety measures [1].

We are not aware of any extensive inventory of active tailings dams worldwide. The lack of comprehensive tailings dam inventories prevents a significant analysis of technical failures, which could serve as a learning tool and help prevent future incidents. Existing records are very incomplete in crucial data elements: design height of dam; design footprint; construction type (e.g., upstream, downstream, and center line); age; design life; construction status; ownership status; capacity; release volume; runout; etc.

Most developed countries adopt frameworks for mining regulation that attempt to mitigate mining disasters whilst respecting community needs, the environment and keeping hazard risks under control [2]. Unfortunately, many developing countries still face a myriad of challenges, such as weak governance, lack of transparency, and lack of accountability in extractive industries, preventing inclusive growth, while threatening both people and the environment. Federal governments’ lack of governance opens the door to informal and illegal mining and the fraudulent or incomplete reporting of activities by mining companies, causing major tax collection losses [3] but also increasing risks to the environment and the population.

Many illegal or informal mining operations take place in and around forests, sometimes overlapping with areas of high conservation value and high watershed stress that are home to vast amounts of biodiversity [4,5]. These mining activities are often difficult to detect from the ground. However, satellite or airborne sensing coupled with geographic information systems (GIS) can be used to assess where hazardous natural or man-made phenomena occur or are likely to occur over a wide area. This information can feed into risk assessments or illegal activities detection, along with information on natural resources, population, and infrastructure. It can be used by governments and communities in order to prevent or stop dangerous or illegal activities but also to design mitigation strategies to reduce vulnerabilities to acceptable levels.

Unfortunately, the use of GIS technologies in complex and country-wide scale analysis is not currently available to all. It requires access to huge remote sensing imagery datasets, robust computation infrastructure, expensive software tools, and teams of dedicated experts to interpret the vast amounts of imagery. Governments of developing countries are not always prepared, or do not have the financial resources, to deploy such a sophisticated infrastructure.

The main objective of this work was to contribute to the control and supervision of mining projects, as well as to the prevention of illegal or dangerous mining. We seek to meet this objective by proposing a scalable and low-cost method for detecting tailings dams and mines in wide areas combined with a risk assessment strategy for the identified mines. The scientific objective was to explore the combination of recent advances in machine learning and freely available multispectral satellite imagery, seeking to propose solutions adapted to a very specific and atypical context, where one wants to identify “objects”, specifically dams and mines, of very different shapes, dimensions, and hue in low resolution imagery covering huge geo-spatial areas.

## 2. Background

New technological trends may help to reduce both the need for significant in-country technical expertise and computation infrastructure. Free cloud computing resources, free open-source software tools and deep learning methods can be used to build inexpensive and powerful data analysis tools. Easy access to these powerful technologies, coupled with the support that can easily be obtained from numerous online technical forums, tutorials, and online training opens new frontiers for their unrestricted use by governments, entities, and communities committed to the social good. Research and training carried out by universities and research centres, often in cooperative projects between different countries, has also contributed to a rapid expansion in the use of these new technologies (This work was made possible through a collaboration between the University of Oxford in the UK and Brazilian researchers). Anyone can access the latest free Earth observation imagery, from optical and radar satellite imagery to weather data and digital elevation maps. Forty years worth of free satellite images can be obtained from USGS-NASA Landsat (https://earthexplorer.usgs.gov/) or European Space Agency (ESA)’s Copernicus Open Access Hub (https://scihub.copernicus.eu/). Google Earth Engine (https://earthengine.google.com/) can be used at no cost for the searching, previewing, and preprocessing of these large free GIS image sets.

The analysis of remote sensing images in combination with machine learning methods has seen a massive increase in popularity in recent years. Machine learning was applied to several tasks, including scene and object classification, object detection, land cover, use and classification (LULC), object-based image segmentation, and analysis (OBIA) [6]. To explore the combination of free cloud computing, free software, and deep learning in analysing a real large-scale problem, in this paper, we demonstrate an application in automatic country-wide identification of surface mines and mining tailings dams in Brazil, followed by the classification of their potential environmental impact. The approach uses open-source and freely available software including Python [6] and TensorFlow [7] that can be run for free on Google cloud infrastructure Colaboratory (or Colab) (https://colab.research.google.com/notebooks/welcome.ipynb). Our paper is accompanied by open-source Python code and data that can be adapted to other Earth observation (EO) applications by government departments and their supervisory bodies and also by local communities and non-governmental organisations. The code supporting this publication can be accessed at: https://github.com/remis/mining-discovery-with-deep-learning. Although this study focuses on mining activity in Brazil, the proposed approach is applicable to any region where satellite imagery is available.

Section 2.1 describes the environmental and social impact of illegal mining and motivates a Brazil case-study. After presenting related work in Section 2.2, we describe the data processing methodology, specifically, our convolutional neural network approach, and its application to mine detection and tailings dam environmental risk assessment in Section 3. The efficacy of our approach is demonstrated on the Brazil case-study in Section 4. Section 5 summarizes the main results obtained in this research. We conclude in Section 6.

### 2.1. Mining Processes and the Brazil Case Study

Surface mining accounts for two-thirds of the world’s metals used in industrial and technological applications and produces geochemically abundant metals such as aluminium, iron, magnesium, coal, manganese, and titanium, and also geochemically scarce metals, which include copper, lead, zinc, gold, and silver. Large-scale quarrying and open-cast mining is also used for stone, sand, gravel, crushed stone, and phosphates extraction. Surface mining usually uses heavy equipment, such as earthmovers, to remove overburden, followed by large machines, such as dragline or bucket wheel excavators, to extract the mineral. Large mining companies typically use appropriate methods and equipment and assume responsibility for environmental damage or impacts. Illegal mining, by contrast, uses makeshift methods and never takes responsibility for the damage it causes. In some rainforest territories, illegal miners use ferries to search the riverbed for precious metals. In others, as well as the rafts, there are almost industrial scale mining fronts, where backhoes and dredgers dig craters in the woods. This is the case for many of the mines in the Tapajós region of Pará, Brazil, where satellite and aerial images show large open “scars” in the forest, even on protected indigenous land (see, for example, Figure 1 showing an aerial photo taken in 2019 by journalists flying over a huge illegal mining area in the Kayapós indigenous reserve, in the state of Pará (https://www.bbc.com/portuguese/brasil-49053678).

Metal mining uses a huge amount of water in the crushing and ore separation process and leaves behind residual rocks that have been reduced to the consistency of gravel or sand, called tailings. Tailings are a mixture of particles suspended in liquid, usually toxic and potentially radioactive, and can be deposited in large ponds. These ponds, called “tailings dams”, are enormous lake-like structures, square-kilometres in size, and are held in place by earth-filled dams. The dams in mountainous valleys can reach as high as 300 m. These dams are designed for long-term containment and are some of the largest man-made structures on Earth.

According to Brazilian legislation, those interested in exploring the subsoil for the extraction of ores must obtain a mineral exploitation concession through the mining concession regime, aimed at the extraction, processing, and commercialization of the mineral asset. Before obtaining this concession, the entrepreneur must follow a series of steps, starting with the request for a research authorization [8].

The Brazilian National Mining Agency (ANM) is responsible for supervising the mining activity in Brazil, from the authorization of mining to supervision of the enterprise, including the collection of taxes from mineral exploration and the inspection of tailings dams. Unfortunately, ANM suffers from a poor administrative structure with insufficient staff numbers, a lack of training, technological deficiencies, and an annual budget that has been cut by successive governments [9]. In 2019, the agency had only 35 inspectors to inspect the 790 ore dams officially declared across the country (https://brasil.estadao.com.br/noticias/geral,pais-tem-apenas-35-fiscais-de-barragem-de-mineracao,70002699885).

ANM’s work on controversial projects, including mining in conservation areas and indigenous lands, is also hampered by legal loopholes. There is no consensus in the normative scope for mineral research in protected areas. The Brazilian federal constitution authorizes preliminary research and mining on indigenous lands, but requires specific regulations from the National Congress, after hearing from the affected communities [10]. As this regulation does not yet exist, instead of rejecting proposals from mining companies, ANM can only attempt to delay them, arguing that they wait for the creation of an appropriate law. Meanwhile, mining companies, some armed with provisional authorizations from the courts, continue their enterprises with impunity.

According to a World Wildlife Fund for Nature (WWF) report [11], in 2018, there were 5675 requests for mining concessions awaiting the approval of the ANM that overlap totally or partially with protected areas within the scope of the Legal Amazon. This demonstrates the magnitude of the threat to these protected areas and the pressure to remove restrictions on mining.

### 2.2. Related Work

Remote sensing imagery has been used for surface mine monitoring for several decades [12]. Recent increased availability and diversity of satellite image sets has sparked a wider range of studies to monitor environmental impact of mining activities using visual or computer-assisted interpretation with automated pattern recognition techniques [13,14,15,16]. This work is, in part, driven by the lack of databases containing geographic coordinates, characteristics, and attributes of mines. These characteristics of mines are important for environmental management and monitoring. The United States Geological Survey (USGS) provides the Major Mineral Deposits of the World (MMDW) database (https://mrdata.usgs.gov/major-deposits/) containing locations of known surface mining areas around the world. Yu et al. [13] used this database to investigate the effectiveness of remote sensing datasets to identify and map land cover changes related to surface mining activities, but reported a lack of precision related to geographic location, geometry and spatial extent of deposits. Soulard et al. [17] employed semi-automated procedures to map the country-wide mining footprint and change detection in US using mine seed points derived from official surveys and datasets. However, to the best of our knowledge, no study has been done on automated discovery and classification of unregistered surface mining sites and tailings dams on wide areas.

Space-borne sensors with a wide range of spatiotemporal, radiometric and spectral resolutions have become valuable data sources for large-scale Earth observation applications. However, data availability from multispectral and hyperspatial sensors introduces new challenges in data mining, processing, backup, and retrieval. The complexity and volume of these datasets require advanced data processing, flexible interfaces, and computational power. Cloud-based platforms such as Google Earth and powerful open source software libraries are creating new opportunities and bringing in new users and applications of remote sensing in environmental sciences and natural resources management. For a survey on processing remote sensing data in cloud computing environments, we refer readers to [18].

Machine learning methods have gained strength with these new resources. Deep learning methods have obtained state-of-the-art results in many Earth observation applications, such as image classification [19], semantic segmentation [20], phenological studies [21], poverty mapping [22], precision agriculture [23], and detection of pivot irrigation systems with very-high spatial and temporal resolution imagery [24]. An extensive survey on deep-learning-driven remote sensing image scene understanding can be found in [25]. In deep learning, convolutional neural networks (CNNs) play a central role for processing visual-related problems [26]. CNNs have been successfully employed, for instance, to classify hyperspectral images directly in the spectral domain [27], pixel-wise classification of satellite imagery [28], and detection of informal settlements in very-high resolution (VHR) satellite images [29].

## 3. Materials and Methods

### 3.1. Data Acquisition

Deep learning methods require rich training datasets to fit, evaluate, and test the neural network models. Preparation of these datasets requires a considerable number of good quality images containing the targets of interest. For applications in remote sensing it is imperative to obtain a representative collection of satellite imagery and address several corresponding technical issues such as cloud cover, select appropriate spectral bands, and undertake image registration. For supervised learning it is also paramount to have trusted sources of ground truth labelled imagery. In this work, we used, as a starting point, two official georeferenced databases released by the Brazilian federal government. The first is a catalogue of 612 officially reported tailings dams in Brazil, made available by the Brazilian National Mining Agency on its website (https://app.anm.gov.br/sigbm/publico). This database contains some information about each dam, including its owner and spatial coordinates of the dam’s location. There is also some technical information including the main ore mined, height, volume, construction method, risk category, and potential damage that could arise if the dam were to fail. The second database is a catalogue of 268 legally operating ore mines in Brazil obtained from the same agency, but is not publicly available.

Unfortunately, both of these databases have quality issues. A significant number of coordinates point to locations near the mines or tailings dams but not exactly to them, or they are completely off target. A visual inspection of each reference using Google Earth images was required to address this problem and correct the target coordinates. Ferreira et al. [30] used the same databases to create the *BrazilDAM*, a public dataset based on Sentinel-2 and Landsat-8 satellite images covering the tailings dams catalogued by the ANM. Once the main database was created with the central coordinates of the mines, work was done to collect points of interest around these locations. Large mines can occupy tens of square kilometres and contain multiple dams. Thus, seeking to increase the number of examples (data augmentation), several coordinates at strategic points of the mining sites were recorded.

Our machine learning classifiers require samples of all target classes to train them. This includes the background class, which are objects other than dams and mines. Thus, it was also necessary to create a database with coordinates of locations where no mines or tailings dams were present. A careful choice of representative locations was necessary in order to capture the diversity of the background class such as cities, construction sites, and natural lakes. Consequently, two databases were created: one containing 1397 coordinates pointing to mines or tailings dams and another containing 1463 background locations. From this set of points, bounding boxes centred on these (latitude, longitude) coordinates defined the sub-imagery used for training the classifier.

As our goal was to implement an approach using only freely available resources that can be used to identify mines and dams in any region of the Brazilian territory, we chose to use only open access satellite imagery. The main providers are NASA and the European Space Agency (ESA), offering the Landsat Mission datasets (https://landsat.gsfc.nasa.gov/data/) and the Copernicus–Sentinel program datasets (https://sentinel.esa.int/web/sentinel/sentinel-data-access), respectively. Landsat satellites provide the longest continuously acquired set of space-based remote sensing data, with Landsat-8 being the latest member of the constellation. The Sentinels are a constellation of satellites deployed by ESA as part of the Copernicus program, which include high-resolution optical images from Sentinel-2A and 2B. The Sentinel-2s are sun-synchronous and multispectral instruments (MSI). The spatial resolution of a remote sensor image is strongly related to the level of detail that can be retrieved from a scene. Image resolution is usually measured using the distance between adjacent pixel centres measured on the ground. The spatial resolution of most bands on Landsat-8 is 30 m whereas the main visible and near-infrared Sentinel-2A bands have a spatial resolution of 10 m. The revisit time of a satellite system (i.e., the time elapsed between subsequent observations of the same area of interest) is another decisive factor of choice. The revisit period of Landsat-8 is 16 days whereas Sentinel-2A’s revisit period is 10 days. Given their better resolution and sampling frequencies we chose to use the Sentinel images. Sentinel band resolutions and wavelengths used in this study are shown in Table 1.

To access and process the huge Sentinel-2 COPERNICUS/S2 image collection, we used the Google Earth Engine platform (GEE). The GEE [31] is a cloud computing platform designed to store and process large spatial and remote sensing data sets for free. The platform offers a friendly and convenient interface for the development of algorithms using JavaScript and an application programming interface (API) with image pre-processing features that allow solving some common problems in geoprocessing, such as cloud cover. In addition to the computing infrastructure, Google also archived all Landsat and Sentinel image data sets and linked them to the cloud computing engine. The GEE Image Collections are stacks of raster images that can be processed as whole on single API calls, making it easy to process complex operations over a selection of images and their bands. This structure allows operations like filtering, mapping, reducing, compositing, and iterating on large image sets. The results can be immediately visualized on screen or exported to a Google Drive folder. The same API can be used from a Python Jupyter Notebook interface on Google Colaboratory.

In order to acquire and prepare the image sets, we followed the workflow presented in Figure 2. In the first step (1), the desired data collection is selected and then date and cloud cover filters are applied. To ensure acquisition of good quality images, a broad time-frame of two years, starting January 2018, was chosen. Using the metadata of the images, it was also possible to select only the images within this time-frame with less than 20 percent cloud cover. In the second step (2), the image collection is processed in order to obtain a single raster image. A pixel based function was applied to the image stack to remove pixels containing cloud or cloud shadow patterns evident in the QA60 bit band. Finally, to define a single raster image a pixel based median operation was applied to the image stack. The median value at a location was the most common valid (i.e., no cloud) pixel value from all images at that location. This operation is based on the assumption that clouds are not in the same place permanently.

For the input to our classification algorithm, the coordinates of the points of interest must be in tabular format, and made available as comma-separated values (csv file) or Esri shapefile (shp). To facilitate the processing of shapefiles, we imported them into GEE as feature collections before loading them into Google Colab as arrays using GeoPandas (https://geopandas.org/) (i.e., workflow step 3). A small square image was extracted around each coordinate centred on the coordinate. Each of these sub-images contains enough information to determine the presence, or not, of a mine or dam. Two sets of image crops were prepared. The first one had 2 × 2 km (0.018 × 0.018 degrees) image extracts, each comprising 201 × 201 pixels and 12 spectral bands. These small image extracts were used to train and validate the mine discovery classifier. The 2 × 2 km bounding box size was as large as possible within the memory limitations of the GPUs available in the Google Colab environment. The second set of cropped images, comprised only the point coordinates for which the mining ore was annotated, had 200 × 200 m (0.0018 × 0.0018 degrees) image extracts, each comprising 21 × 21 pixels and 12 spectral bands. In this case, the 200 × 200 m bounding box size was chosen in order to focus on specific locations in the mine where it was possible to identify excavations, water, or ore residues exposed on the surface. This second dataset was used to train the environmental impact classification model. Figure 3 shows an example of a mine and its tailings dam and Figure 4 shows the same spot on a cropped image from the Sentinel-2 collection obtained using the workflow in Figure 2.

### 3.2. Deep Learning

Deep learning methods have obtained an unprecedented success in computer vision. High accuracy has been achieved on complex tasks such as the ImageNet Large Scale Visual Recognition Challenge (ILSVRC) to evaluate algorithms for object detection and image classification at large scale [32]; however, several aspects need to be taken into consideration when applying these methods in real life cases. State-of-the-art computer vision algorithms use thousands or sometimes millions of images with known annotations for training and evaluation on large scale problems. For real life cases this can be unfeasible due to the lack of examples or dubious interpretations of the object classes present in an image. A level of automation classification accuracy is also task-specific in real life applications. In applications where a classification mistake has a high cost and may cause harm, inaccuracies inherent in machine learning methods may not be tolerated. In applications where automated classification methods are used only to support expert decisions, a small number of false positive or false negative classifications may be tolerated. In this research, which looks for indications of the presence of surface mines or tailings dams as well as of the type of associated environmental impact they have, it would perhaps be impracticable to obtain a highly accurate system given the limited amount of training examples and the numerous variations in shape, hue, areas, and volumes of different mines and ore categories. Thus, we designed a complete analysis strategy to support the analyst in the discovery and classification of mines and dams in the most diverse regions of the planet, prioritizing the reduction of false negatives, so as not to lose relevant indications, and seeking to maintain false positives at acceptable levels.

#### 3.2.1. The Convolutional Neural Network

Advances in deep learning in computer vision have been strongly driven by convolutional neural networks (CNNs) in recent years [33]. CNNs evolved from multilayer perceptrons, which are networks consisting of an input and an output layer, as well as multiple hidden layers, and use a stochastic gradient decent based optimization algorithm to learn their parameters efficiently. A Neural network is defined by its layer architecture and activation functions. A CNN typically comprises a series of convolutional layers, each performing local “filter” operations that detect features in the output of previous layers. The neural net learns the filters that, in traditional computer vision algorithms, are hand-engineered. Other types of layers are also used in CNNs such as pooling, fully connected, and regularization layers. Pooling layers combine the outputs of groups of neurons in the previous layer in order to reduce data dimensionality. A fully-connected layer connects all neurons in one layer to all neurons of the next layer. They usually appear after the convolutional layers so that the fully-connected layers capture complex relationships between high-level features extracted by the convolutional layers. CNNs may be prone to overfitting. To mitigate this undesired effect, they are usually supplied with regularization layers. One method to reduce overfitting is *dropout* where, at each training cycle, the outputs of some nodes are ignored with a given probability to prevent the nodes from co-adapting to the data. The activation function is responsible for transforming the weighted sum of a node’s input into that node’s output value. The activation function commonly used in CNNs is the Rectified Linear Unit (RELU). For a review of the main concepts related to deep learning please refer to [34].

#### 3.2.2. Visual Pattern Mining

We explore the application of CNNs on two related environmental problems: the discovery of surface mines with their corresponding tailings dams in raw satellite imagery covering large areas (described in Section 3.3) and the classification of the environmental impact of the discovered mines and dams (Section 3.4). These problems required some further adaptation of the CNN methodology, which we describe next.

Computer vision problems are usually categorized as *image classification* when the goal is to classify the whole image, as *object detection* when the goal is to detect and classify one or more objects within an image displaying their position with bounding boxes, and *semantic segmentation* when the goal is to recognize the object class for each pixel of an image. These models traditionally require training data in the form of classified images, objects encased in bounding boxes or highlighted by masks. However, we use, as input, only point valued data—the spatial coordinates of the dam locations. We do not have more informative masks nor bounding boxes available for image segmentation nor object detection, preventing us from using models such as the Unet [35] or region-based CNNs [36] for example. As Mines and dams frequently do not have clear edges to delimit them in the imagery (see, for example, Figure 5 and Figure 6) and their sizes and shapes vary enormously we cannot easily create bounding boxes nor masks to augment the training data.

An alternative machine learning approach to computer vision problems is visual pattern mining (VPM) [37] which we can adapt for point labelled data. This paper demonstrates the VPM approach has sufficient accuracy for dam localisation using point labelled data. In essence, VPM classifies parts of an object. VPM is often implemented as a two phase approach. Individual parts of an object are classified in Phase 1 and then Phase 2 classifies the object by using spatial relationships of the parts identified in Phase 1. In Phase 2 the classification of visual patterns is obtained from the combination of multiple filters from the final convolutional layer of the Phase 1 CNN. Visual Pattern Mining is widely used for mid-level feature representations on image classification tasks and visual summarisation. VPM code is freely available via the Python implementation PatternNet (https://github.com/KnurpsBram/PyTorch-PatternNet).

We adapted the pattern mining method to both mine and dam identification in satellite imagery as well as the classification of different types of construction and the corresponding environmental impact. Our adaptation uses Phase 1 of the VPM approach to identify parts of dams or mines. We do not distinguish the nature of the individual parts themselves, only that a sub-image of interest either contains or does not contain a part of a dam or mine. This allows us to fix the size of the CNN input image, and thus provides us with an efficient classifier that scales over large areas of interest. The new algorithm was trained on small image samples, some of which contained a part of a dam or mine, annotated with the presence or not of a mine or tailings dam and the ore type if it was known. The data acquisition process was detailed in Section 3.1. Each image sample covers a 2 × 2 km area (201 × 201 pixels). This size was chosen after several modelling cycles where we tried to find a compromise between the prediction performance and the GPU’s memory capacity.

We implemented the VPM method as a fully convolutional network (FCN) for computational efficiency and scalability. Since an FCN contains only convolution and pooling layers, with fully-connected layers converted to convolution layers, it can be executed efficiently on variable size images. This is particularly useful for our research problems, where training needs to be done on small image patches, showing parts of a mine or a dam, but classification is performed on big images covering wide areas of Brazil. When applied to wide areas, output of the FCN is a map comprising clusters of class markings, each cluster marking a dam or mine.

The FCN algorithm was implemented using open-sourced computational resources, which are freely available and easily accessible online. CNN-based applications can be developed and deployed using open source libraries such as the Keras TensorFlow’s high-level application programming interface (API), used for building and training deep learning models [7]. It is useful for fast prototyping, state-of-the-art research, and production. The Keras TensorFlow API is available on Google Colaboratory (Colab), a free cloud service based on Jupyter Notebooks that offers robust graphics processing unit (GPU) based computing at no cost [38]. On Google Colab, a developer has access to a flexible runtime fully configured for deep learning, avoiding all the workload required to prepare the infrastructure and environment that machine learning projects typically require. Large files can be uploaded, accessed and saved using Google Drive, greatly simplifying data input–output operations. The existence of such a platform both simplifies and reduces the cost of developing and running sophisticated machine learning methods, including deep learning, thus allowing its use for social good by governmental and non-governmental organizations in developing countries and not just by conventional academic and business organisations.

### 3.3. Mine and Tailings Dam Discovery

Our FCN architecture comprises five convolution layers, three pooling layers, and a dropout layer with a 50% dropout rate for regularization. The model contains 259,810 parameters, all trainable and distributed on the network layers, as presented in Table 2. As shown in the table, the 10 layers are arranged in sequence, with the input of images occurring from level 1 and the prediction response at level 10. We used Adam stochastic optimisation [39] to iteratively update the network weights, and minimise the categorical cross-entropy loss function, due to its computational efficiency and little memory requirements. All convolution layers use the rectified linear unit (RELU) activation function and are initialised with a normal weight distribution. The input training data are composed of 201 × 201 × 12 float arrays. They correspond to the 12 spectral bands of 2 × 2 km raw Sentinel-2 image samples. These images are normalised with each band having zero mean and unit standard deviation across the entire training data. Only the number of bands in the imagery is fixed (i.e., 12) in the FCN input layer, the width and height of the input images are not specified to allow different sized images for training and classification. The output layer is a convolution with a 1 × 1 kernel and two filters, corresponding to two target classes: *mine* and *not mine*. The *mine* class corresponds to scenes containing surface mines and/or tailings dams.

Our FCN architecture is rather conventional and the implementation of its training, validation, and testing procedures was straightforward on the Keras TensorFlow API. The fact that the input data consists of raster images, stored as TIFF (Tagged Image File Format) files, requires proper load and translation steps in order to prepare the float arrays (or tensors) required as input to the CNN. We used the Geospatial Data Abstraction Library (GDAL) (https://gdal.org/) to implement these steps. A total of 2860 201 × 201 × 12 images were used to train and test the FCN.

### 3.4. Environmental Impact Classification

Various principles and processes are utilised within mining that determine the impact they have on the environment [40]:Low environmental impact: extraction and any processing of sand, gravel, or rock where the environmental impact is usually restricted to dust or noise. It is usually related to extraction for construction purposes or as raw materials for industry. The processing does not go beyond crushing or screening of sand and gravel.High environmental impact: extraction of minerals and ores in facilities, which produce large amounts of waste consisting of finely crushed rock (tailings) and discharges of polluted waste water. This category comprises mining sites where the valuable substances in the ores must undergo processing and where large amounts of tailings must be deposited. Within this category, a sector of great environmental importance is the mining of sulphide ores containing copper, zinc, lead, or nickel, often in combination with pyrite. These ores are the most common ores on a global scale. The environmental impact of this type of mining can last a long time after the end of activities (An exception to this simplified classification of environmental impact concerns gold mines. Although its process is similar to that used in the extraction of gravel, based on washing, crushing, and screening, the environmental impact can be great due to the damage it can cause to water courses and the use of products that are harmful to the environment such as mercury; however, in low resolution satellite images it is not possible to distinguish a gold mine from a simple exploration of sand or gravel.)

In order to provide an indication of the potential environmental impact of a discovered mine or tailings dam we explored again the usefulness of deep learning based classifiers. Using the Brazilian officially declared mines and tailings dams databases, containing the ore type declared by their owners, we chose 1275 representative point coordinates and acquired Sentinel-2 image patches, using the protocol described in Section 3.1. Each image patch has 21 × 21 pixels and 12 spectral bands. The classes considered in this second predictive model, their related materials and the number of image samples used as the training dataset are:high impact waste: lead, iron, niobium, zinc, copper, phosphate, nickel, schist, manganese, and bauxite; 425 samples.low impact waste: sand, gravel, clay, and stones; 425 samples.no ore: soil, rocks, and vegetation; 425 samples.

This second model used a conventional CNN network architecture, consisting of two convolution layers followed by a max pooling layer each and two dense layers. The input is an image patch of a dam or mine and the output is an one-hot encoding of three classes: high, low impact, and no-ore. Due to the smaller 21 × 21 training image patches the convolution layers used a reduced convolutional filter (kernel) of size 2 × 2. The max pooling layers also used a 2 × 2 kernel. The first dense layer has 1024 elements. The second has 3 elements, corresponding to the output classes, and uses a softmax activation. The architecture, comprising 2,373,219 parameters, all trainable, is presented in Table 3, where the 9 layers are arranged in sequence, with the input of images occurring from level 1 and the prediction response at level 9. Adam stochastic optimisation is used to minimise the softmax cross-entropy function.

## 4. Experiments

It is important to note that the content presented here summarize the most significant results obtained after a long process of attempts, with several sets of images, modelling strategies, and CNN architectures implemented and tested. It would not be feasible to show here details of all the studied configurations and comparisons of all the intermediate results obtained.

### 4.1. Validating the Mine and Dam Localization Model

The FCN described in Section 3.3 takes approximately 100 seconds to train on Colab GPUs. Figure 7 shows the iterative evolution of the model accuracy on ten re-sampling, training, and validation procedures (k-fold cross validation). We obtained an average accuracy on the training data of 99.23±0.7% after 30 epochs. In each re-sampling step, the FCN was validated on a separate dataset containing 286 201 × 201 × 12 images. The mean validation accuracy, F1 score, precision, recall, area under the receiver operating characteristic curve (ROC AUC), and Cohen’s kappa [41] are shown in Table 4. The mean confusion matrix is presented in Table 5.

### 4.2. Validating the Environmental Impact Classification Model

Training the CNN described in Section 3.4 over 30 epochs takes on average 27 s on Colab GPUs. Figure 8 shows the iterative evolution of the model accuracy. The mean validation accuracy and Cohen’s kappa are shown in Table 6. The mean confusion matrix is presented in Table 7.

### 4.3. Wide-Area Maps of Dams and Mines

Google Earth Engine (GEE) provides processed image mosaics for large spatial areas (our paper is accompanied by open-source Python scripts for Sentinel 2 imagery processing on GEE) over which we wish to search for dams and mines. GEE breaks the processed mosaics into a number of GeoTIFF files that can be saved on Google Drive. Each file covers an area up to 92 × 92 km (9216 × 9216 pixels and 12 bands). The volume of those files, around 3 GB, when used as input for FCN classification, exceeds the GPU memory available on Colab. To deal with this limitation, we split each image into four near equally sized patches of approximately 46 × 46 km (2116 square kilometres) each. The FCN output for each patch is a 164 × 164 grid of probability distributions, representing the probability of each class on a certain region of the input image patch.

For each grid point predicted as belonging to the *mine* class we also classify its environmental impact (i.e., low, high, or no ore). The environmental impact CNN uses as input a patch with 21 × 21 pixels (corresponding to an area of 210 × 210 m) from the same mosaic taken around the location of the discovered mine. The CNN output is a probability distribution, representing the probabilities of each environmental impact class on that location. If the *no ore* class has the highest probability the predicted mine spot is considered a false positive and discarded. To determine the geographic coordinates of the detected mines from the FCN output map, we used Rasterio (https://rasterio.readthedocs.io/en/latest/), a Python API based on Numpy N-dimensional arrays and GeoJSON, to compute the coordinates of discovered targets.

The locations of the mines and tailings dams are exported as georeferenced delimited text files (csv and kml), which can be scrutinized using GIS software or Google Earth. We suggest an analyst visually inspect the CNN discovered mines and tailings dams against the raw satellite imagery or against a base map, like Google Earth’s, to validate the discovered mines or tailings dams. Figure 9 shows the predicted indications of dams and mines pin pointed on a Google Earth base map. It is necessary to understand that each yellow pin points to the centre of an implicit area where a mine or dam pattern was found. In this way, some marks are found in the vicinity of a mine or dam and not exactly in it because the pattern found in that specific prediction could be outside the centre of the considered area. This does not mean that they are false positives. In this way, each mine or dam will be identified by a cloud of nearby marks indicating the region where the mine is located.

We applied our approach to two large areas on the Brazilian territory searching for mines and dams and classifying their potential environmental impact. These areas are indicated in Figure 10. The first area covering 67,110 square-kilometres is located in the state of Minas Gerais. It is one of the areas with the highest concentration of mines and tailings dams in Brazil and includes the cities of Brumadinho and Mariana, where the cited tragic dam collapses occurred. The second area covers 170,267 square-kilometres and it is located in the state of Para where protected indigenous land and environment conservation units struggle against illegal mining and deforestation activities. It includes the Kayapo’s protected land depicted in Figure 1. Both areas are represented by red squares in Figure 10 and the coordinates of the officially registered tailings dams from the Brazilian Agência Nacional de Águas (ANA) database are shown by orange dots. Raster images from both areas were acquired and processed using the workflow in Figure 2. In the “local samples” decision, we follow the “no” path where the areas are defined directly as geo-polygons on step (4) before the crop and export.

The FCN model was run on the 237,000 square kilometres of satellite imagery of Brazil and was able to identify all visible mines on those areas, even small ones of less than one square kilometre in extent. However, unlike the tests performed on the initial collection of examples, a considerable number of false positives were also present in this large-scale prediction. An example of the typical output obtained using the discovery model is shown in Figure 11. This green rectangle covers a 5624 square kilometre area of intense mining activity in Minas Gerais, Brazil. It includes the cities of Brumadinho and Mariana where the recent dam collapses occurred. Yellow and red dots show predicted mines. Blue ellipses indicate where the predictions were confirmed by visual inspection. All other dots are false positives. Two true positives from this area are shown in Figure 12 and Figure 13. Figure 12 shows the large mine covering 46 square kilometres appearing on the top right corner of Figure 11 and Figure 13 shows a small one of only 0.5 square kilometres.

False positives occur in some specific situations. Two common situations are water supply dams, depicted in Figure 14, and river banks shown in Figure 15. Feeding these false positives back into the training data and retraining the network could reduce the false-positive rate. However, as the false-positive rate was acceptable it was our choice to correct these maps manually.

Visual inspection of the positive predictions for the area in Minas Gerais confirmed the discovery of 154 point clusters consistent with mines and/or tailings dams. Sixty three of these clusters are newly discovered mines, not indicated on the official databases of the Brazilian government and not having an official mining concession. Visual inspection for the second area in the state of Para also confirmed a number of relevant positive predictions. Two hundred large unregistered gold mining areas were identified, many of them inside and around protected indigenous land, as can be seen in Figure 16, Figure 17 and Figure 18 (Part of the discovered mines are in areas where a mining authorization was requested but the concession had not been granted, which makes them illegal mines.) Large iron-ore mines were also detected inside two different environment conservation areas.

## 5. Discussion

Using the mine and dams detection model to find potential mining sites, and then applying the environment risk model to the mine images enabled the classification of the environment as shown in, for example, Figure 19 and Figure 20, where two previously known mining sites were identified and correctly classified according to their environment impact. Figure 19 shows a high impact tailings dam in an iron mine and Figure 20 shows a low impact gravel quarry.

We have demonstrated that mines can be detected over a wide area using free low resolution satellite imagery and automation. Although the majority of mines and dams are visible in images of good resolution, and can therefore be identified by a specialist without the aid of an algorithm, it is unlikely that this specialist will be able to do it in a cost effective way in very large areas. What is more, they will need high-resolution imagery, which is either paid for or usually out of date when it becomes freely available. Figure 21 and Figure 22 show that some mines discovered at the Kayapo’s reserve cannot be seen on Google Earth’s outdated imagery.

The limited number of confirmed locations of mines and dams that we have also limits the number and diversity of examples we have for training the CNNs, which poses a risk of overfitting. Data augmentation strategies were used, but due to the enormous diversity of format, scale, and shades of colour that a mine or dam can have, the number of images used is still modest. Due to the difficulty in expanding this dataset, we opted for a hybrid strategy of validation of the predictive models. Several CNN architectures were explored and the testing and validation process was carried out both on images from the dataset itself but also in the large areas already described. These experiments in large areas effectively showed results below those obtained in the training and test bases, mainly with regard to false positives, but they also proved their potential to discover new mines and dams, confirmed by visual inspection.

The two-stage prediction strategy also strengthens the method. The first model, which predicts the location of mines and dams, generates inputs for the second model, which predicts the environmental impact of the ore explored there. This second prediction stage serves as a filter for false positives when no ore is identified at the indicated location.

We tested the use of pre-trained CNNs, but with results below those obtained by training them from scratch. This is mainly due to the fact that we used images with 12 spectral bands as input and not conventional RGB images. In addition, as already discussed, the visual patterns associated with mines and dams are very specific, not consistent with those found in the conventional image sets used to train the CNNs available for transfer learning.

## 6. Conclusions

Our work has found 263 tailings dams hitherto unknown to the Brazilian government, distributed across huge areas of the country; a task that would be very time consuming, prone to error, and costly if performed manually.

Despite the limited number of available training examples and the enormous diversity of visual patterns that mines and dams can present, we demonstrated that mines and dams can be identified and their environmental risk classified automatically with an accuracy of over 95% using deep learning, free moderate resolution satellite images, and free computing infrastructure in the cloud.

We developed two deep learning models: one to identify mines and dams and the other to classify their environmental risk potential. Mines and dams occupy a tiny fraction of the satellite imagery. Thus, training was performed on small image patches, containing exemplar mines and dams, and also examples of ’background’ imagery patches. To speed up the identification of dams in wide areas of interest, the trained CNN was expressed as an FCN in the usual way. The environmental risk classification CNN model operates on the mines and dams identified via the FCN. We demonstrated that a conventional CNN is perfectly adequate for this task.

We have integrated our deep learning models into a pipeline for processing multispectral satellite imagery over large geo-spatial areas and provide open-source Python scripts and image datasets, as well as models already trained and ready to be used on new imagery. Our models were trained with images from mines and tailings dams in Brazil, which certainly creates a bias given the local characteristics of the soil, vegetation, and the mining processes used there. The user can choose between applying the models as they are on their own data, retrain the CNNs from scratch using new imagery of mines and tailings dams, or improve the models using additional training with new images (i.e., transfer learning). Our code is publicly available at: https://github.com/remis/mining-discovery-with-deep-learning.

## Figures and Tables

**Figure 1 sensors-20-06936-f001:**
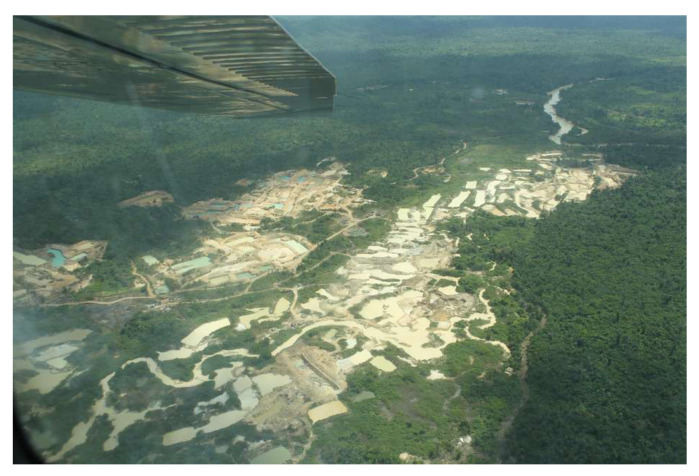
Illegal mining fronts in Kayapó Indigenous Land, Pará. Photo: Ibama/BBC News Brasil.

**Figure 2 sensors-20-06936-f002:**
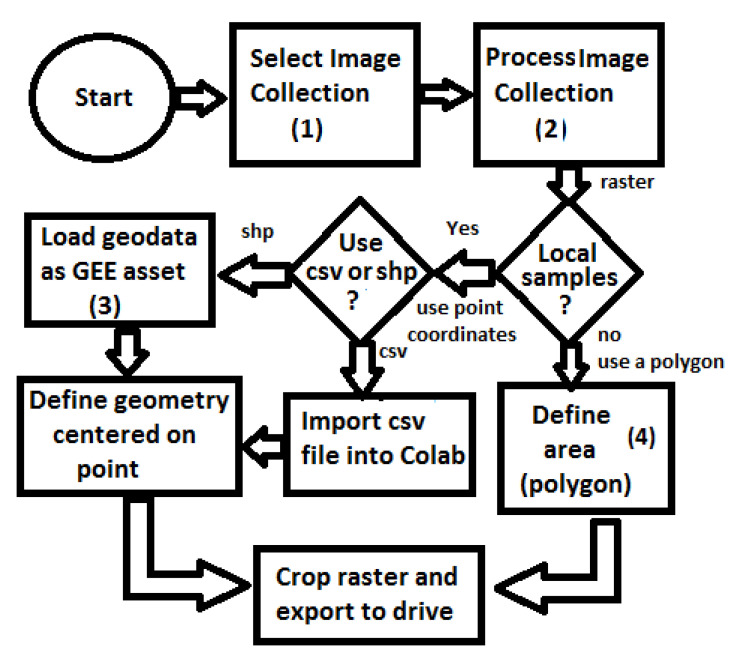
Workflow for accessing and processing Sentinel 2 images.

**Figure 3 sensors-20-06936-f003:**
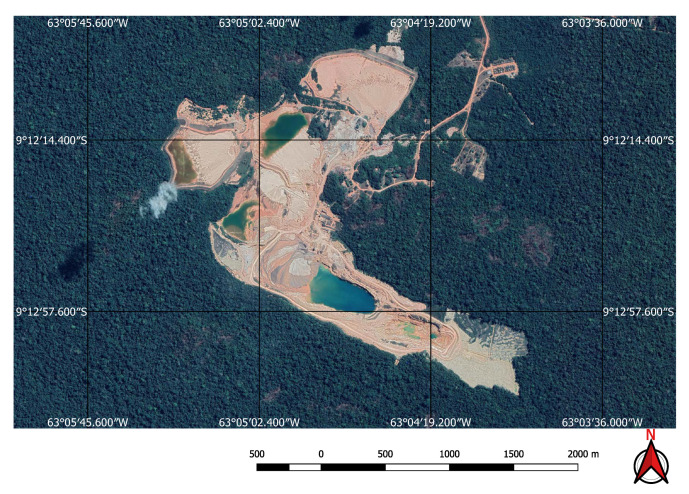
High resolution Google Earth image showing mine and associated dam.

**Figure 4 sensors-20-06936-f004:**
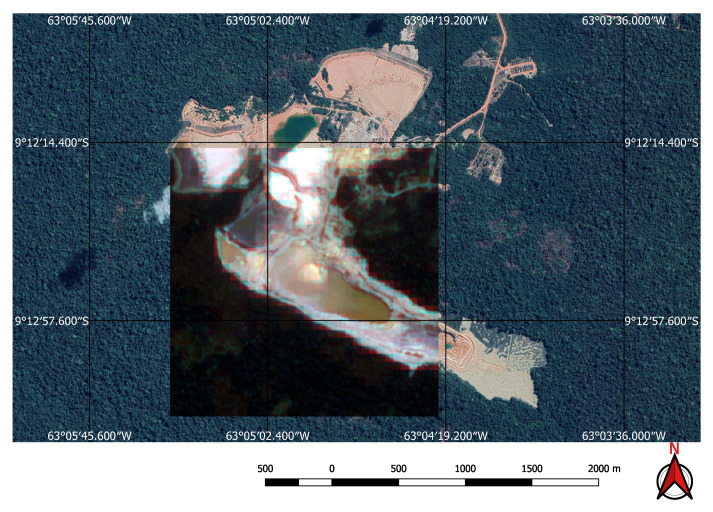
Sentinel 2 image of the mine and dam shown in Figure 3.

**Figure 5 sensors-20-06936-f005:**
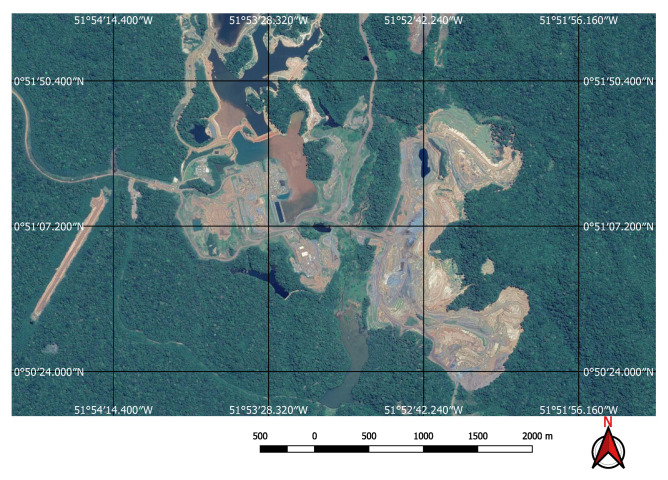
Example of a gold mine.

**Figure 6 sensors-20-06936-f006:**
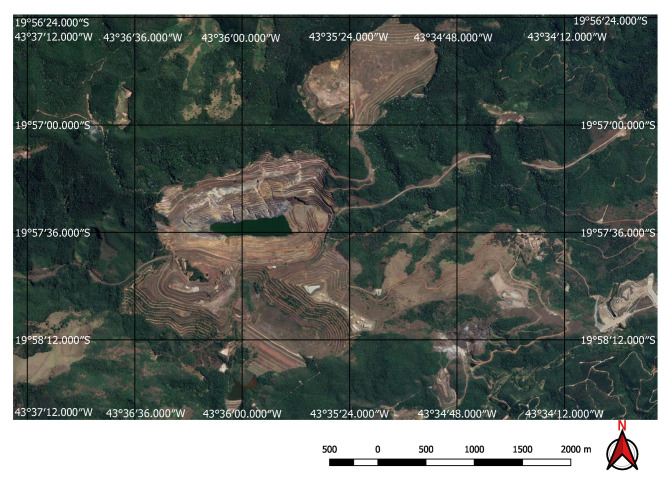
Example of an iron mine.

**Figure 7 sensors-20-06936-f007:**
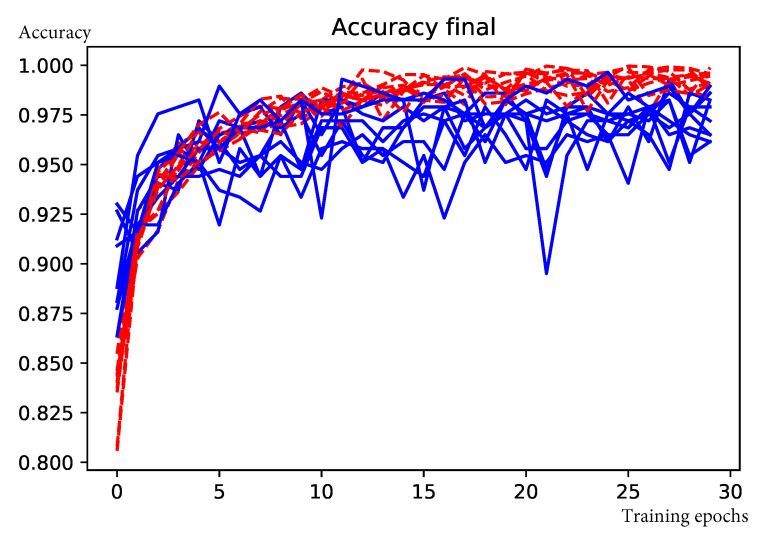
Evolution of accuracy for mines and dams detection over 30 training epochs and 10-fold cross validation. Red lines indicate training accuracy and blue lines validation accuracy.

**Figure 8 sensors-20-06936-f008:**
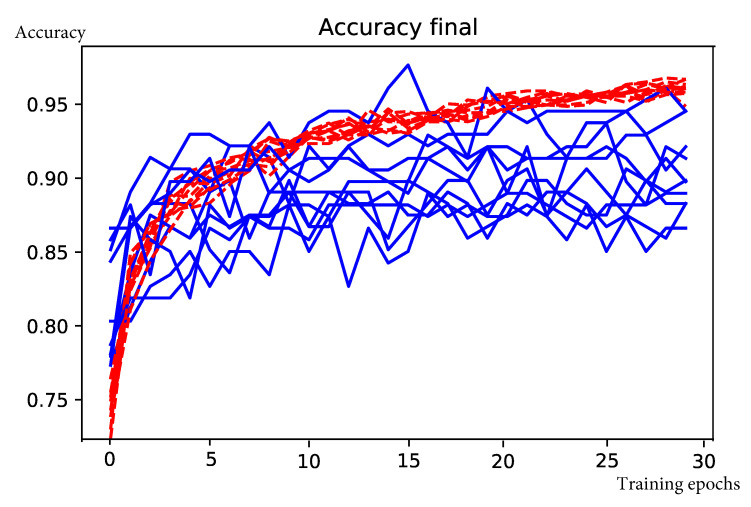
Evolution of accuracy for environmental impact prediction over 30 training epochs and 10-fold cross validation. Red lines indicate training accuracy and blue lines indicate validation accuracy.

**Figure 9 sensors-20-06936-f009:**
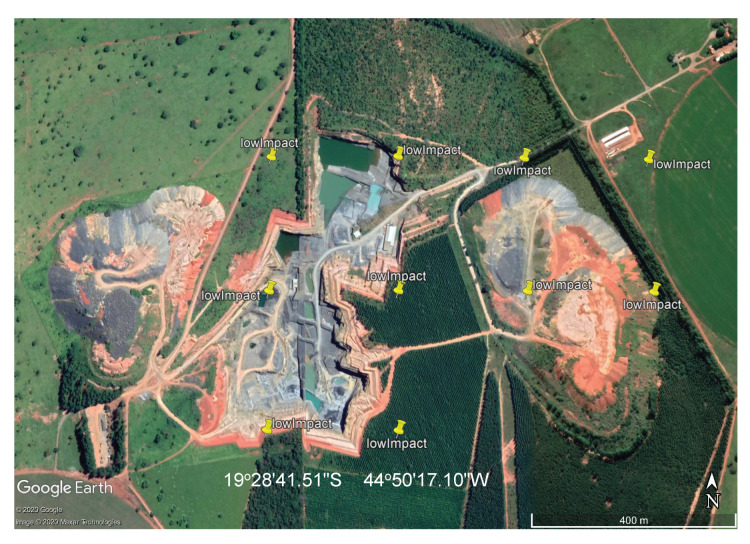
Mines and tailings dams indicated by yellow marks on Google Earth.

**Figure 10 sensors-20-06936-f010:**
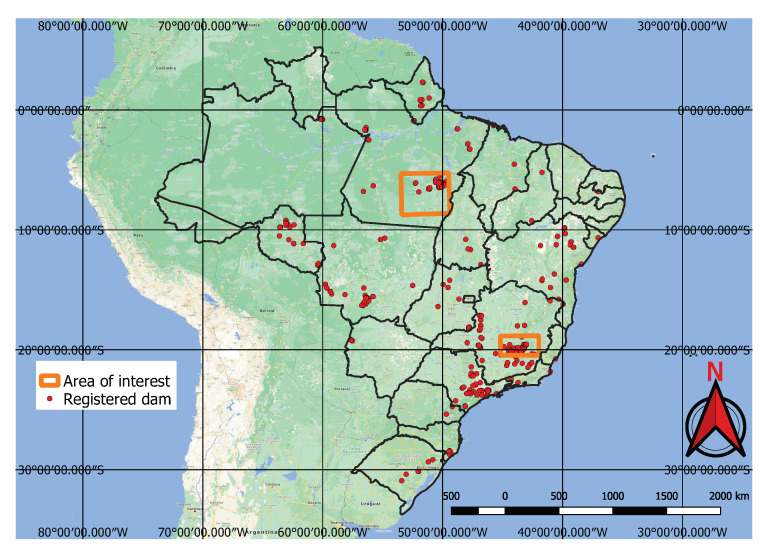
Areas of interest: Minas Gerais (south) and Para (north).

**Figure 11 sensors-20-06936-f011:**
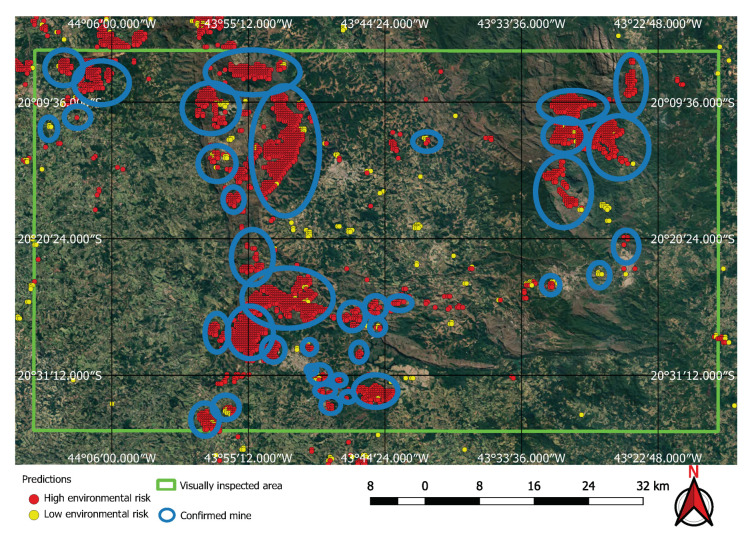
Visual inspection of neural network predictions.

**Figure 12 sensors-20-06936-f012:**
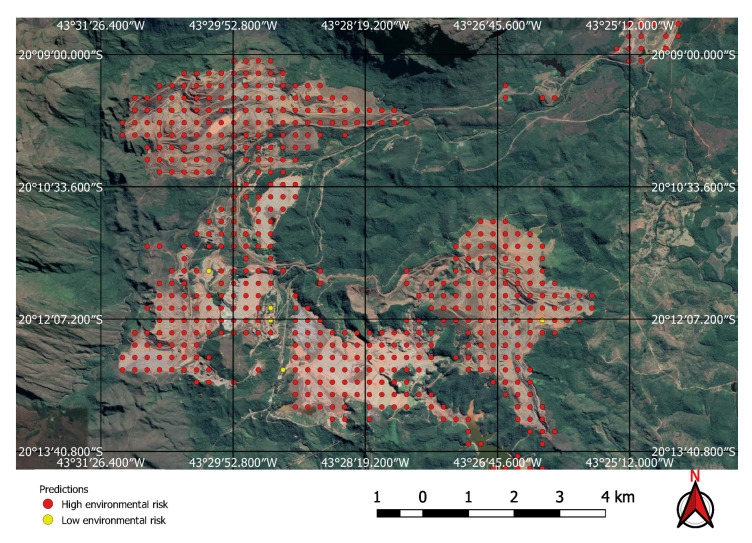
Large mine with potential high environmental impact.

**Figure 13 sensors-20-06936-f013:**
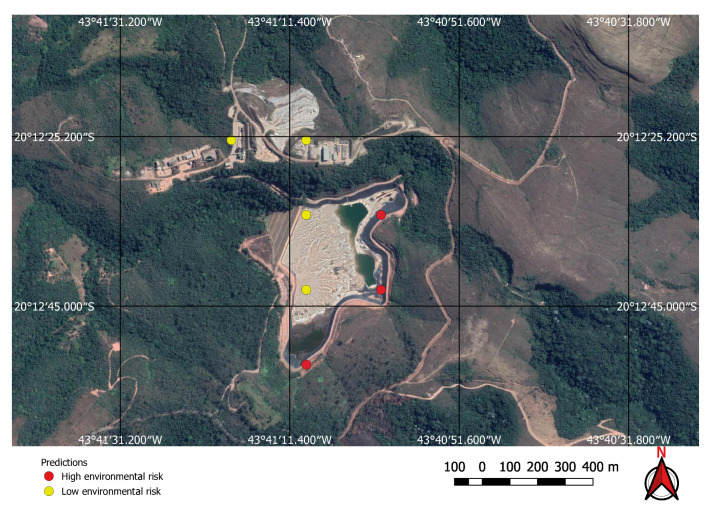
Small mine with potential low environmental impact.

**Figure 14 sensors-20-06936-f014:**
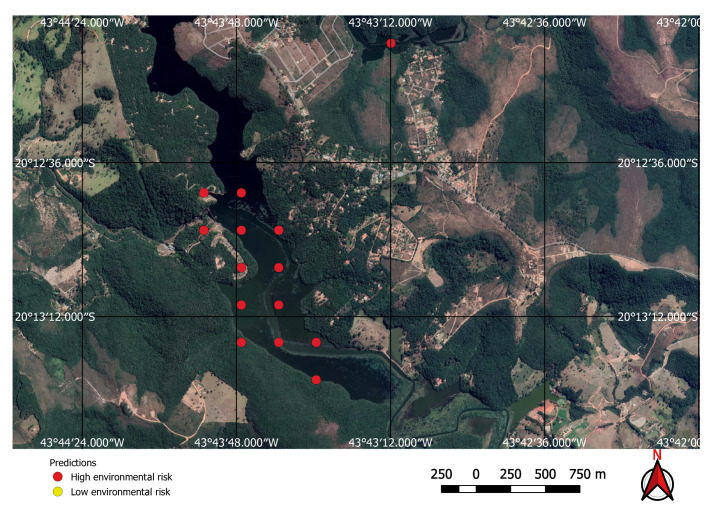
Water supply dam incorrectly identified as a mine by the neural network.

**Figure 15 sensors-20-06936-f015:**
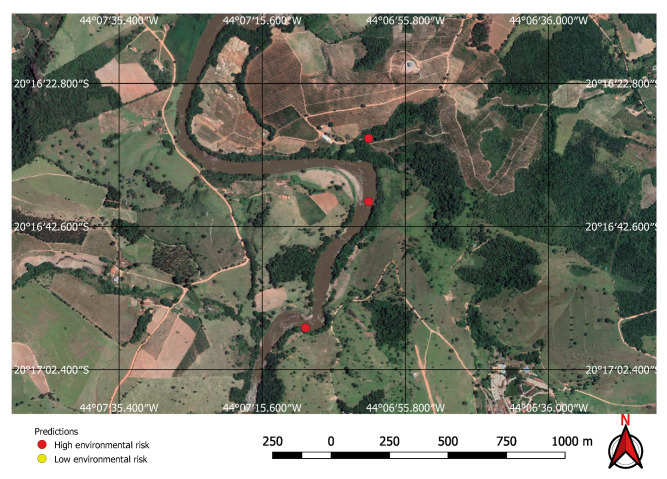
River bank incorrectly identified as a mine by the neural network.

**Figure 16 sensors-20-06936-f016:**
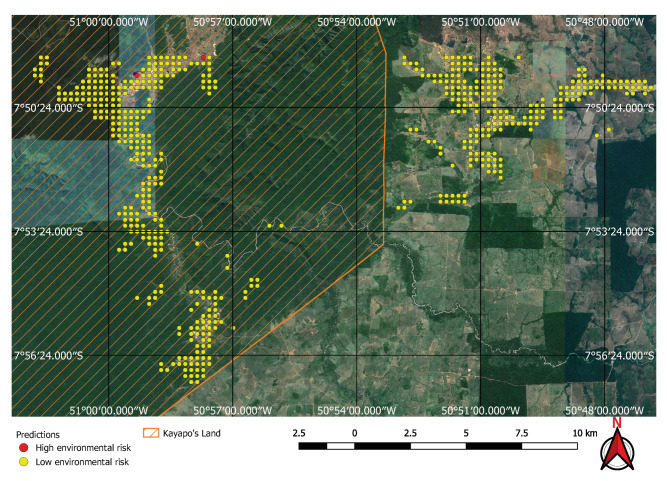
Mines identified on the Kayapo’s protected land.

**Figure 17 sensors-20-06936-f017:**
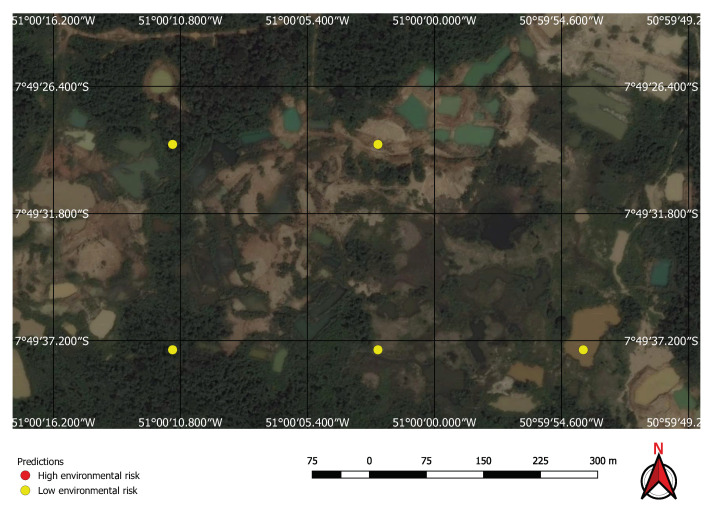
Detail of a discovered mining spot on the Kayapo’s protected land.

**Figure 18 sensors-20-06936-f018:**
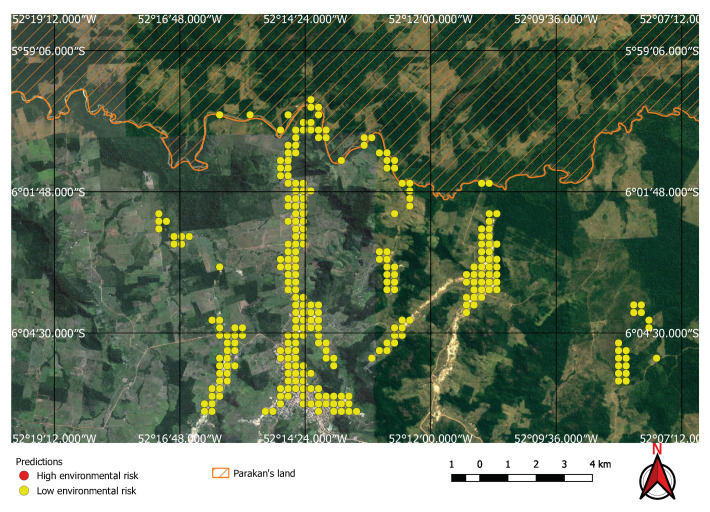
Mining identified by the neural network close to the Parakan’s protected land.

**Figure 19 sensors-20-06936-f019:**
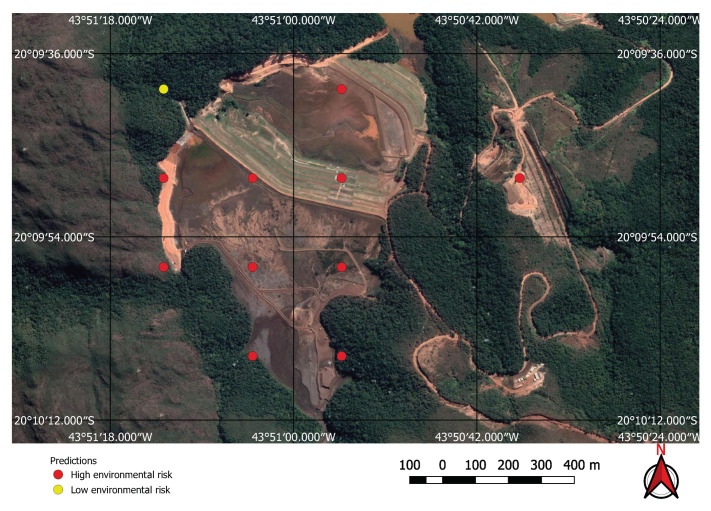
High environmental impact: iron mine tailings dam.

**Figure 20 sensors-20-06936-f020:**
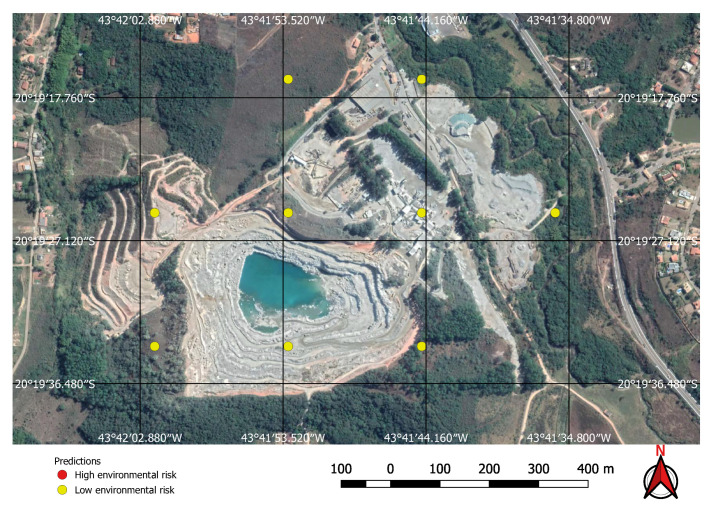
Low environmental risk gravel quarry correctly identified by the neural network.

**Figure 21 sensors-20-06936-f021:**
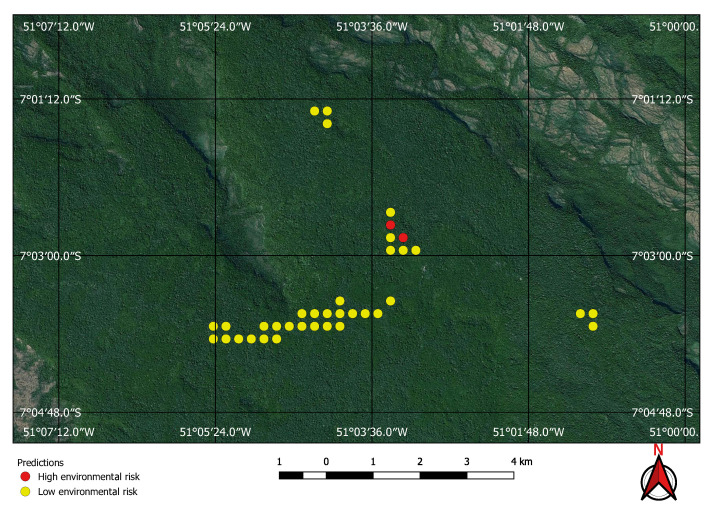
True positives on the Kayapo’s protected land not visible on Google Earth.

**Figure 22 sensors-20-06936-f022:**
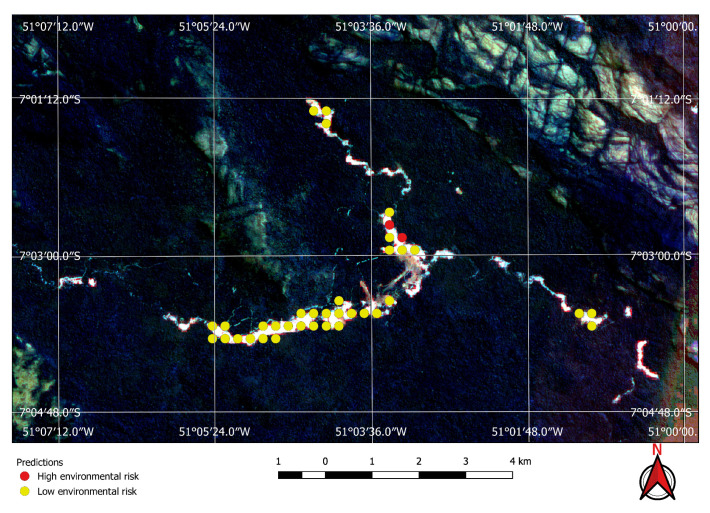
True positives on Kayapo’s protected land visible on Sentinel 2 images.

**Table 1 sensors-20-06936-t001:** The used spectral bands of Sentinel-2.

Bands	Resolution m	Wavelength μm
B01-Aerosols	60	0.44
B02-Blue	10	0.50
B03-Green	10	0.56
B04-Red	10	0.66
B05-Red edge 1	20	0.70
B06-Red edge 2	20	0.74
B07-Red edge 3	20	0.78
B08-NIR	10	0.83
B09-Water vapor	60	0.94
B10-Cirrus	60	1.37
B11-SWIR 1	20	1.61
B12-SWIR 2	20	2.20

**Table 2 sensors-20-06936-t002:** Mine and dam discovery model summary (# means quantity).

Layer	Name	Type	Filters#	Kernel	Param#
1	input	InputLayer			0
2	conv2d_1	Convolution	32	3 × 3	3488
3	maxpooling1	MaxPooling		3 × 3	0
4	conv2d_2	Convolution	64	3 × 3	18,496
5	maxpooling2	MaxPooling		3 × 3	0
6	conv2d_3	Convolution	64	3 × 3	36,928
7	maxpooling3	MaxPooling		3 × 3	0
8	conv2d_4	Convolution	64	7 × 7	200,768
9	dropout	Dropout 50%			0
10	conv2d_5	Convolution	2	1 × 1	130

**Table 3 sensors-20-06936-t003:** Environmental impact model summary (# means quantity).

Layer	Name	Type	Filters#	Kernel	Param#
1	input	InputLayer			0
2	conv2d_1	Convolution	32	2 × 2	1568
3	maxpooling1	MaxPooling		2 × 2	0
4	conv2d_2	Convolution	64	2 × 2	8256
5	maxpooling2	MaxPooling		2 × 2	0
6	flatten	Flatten			0
7	dense_1	Dense	1024		2,360,320
8	dropout	Dropout 50%			0
9	dense_2	Dense	3		3075

**Table 4 sensors-20-06936-t004:** Efficacy of the mine and dam localization model for 10-fold cross validation.

Score	Mean
mean validation accuracy	97.44±1.3%
mean F1 score	0.9749
mean recall	0.9747
mean AUC	0.9746
mean Cohen’s kappa	0.9488

**Table 5 sensors-20-06936-t005:** Mean confusion matrix of the mine and dam localization model for 10-fold cross validation.

Classes	Mine (Predicted)	Not Mine (Predicted)
mine (actual)	136.1	3.6
not mine (actual)	3.7	142.6

**Table 6 sensors-20-06936-t006:** Efficacy of the environmental impact classification model for 10-fold cross validation.

Score	Mean
mean validation accuracy	90.42±3.5%
mean Cohen’s kappa	0.8550

**Table 7 sensors-20-06936-t007:** Mean confusion matrix of the environmental impact classification model for 10-fold cross validation.

Classes	High Impact (Predicted)	Low Impact (Predicted)	No Ore (Predicted)
high impact (actual)	40.7	1.3	0.5
low impact (actual)	2.2	35.9	4.4
no ore (actual)	1.1	2.7	38.7

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
