# Peer review of "Mining and Tailings Dam Detection in Satellite Imagery Using Deep Learning"

_sensors, 2020, doi:10.3390/s20236936_

Round 1
Reviewer 1 Report
Dear Editor,
I have well-received the review request for “Mining and Tailings Dam Detection In Satellite Imagery Using Deep Learning” and would like to thank you for giving me this opportunity to review this interesting article. This paper presents the combination of free cloud computing, free open-source software, shows the automatic countrywide identification and classification of surface mines and mining tailings dams in Brazil.
This is undeniable an interesting paper as the authors attempt to conduct a contribution of low cost data science tools that have high social impact. Their ideas are clearly and well presented in this article.
I would also like to make the following suggestions:
- Some figures are not clear (e.g. Figure 1). It’s better to indicate the location of study area. And shows the orientation of a picture with letters (like E-W) inside the picture than with N arrows. They are much too early in the text, sometimes several pages before the mention in the text.
- The author mentioned two data sets, namely Landsat Mission data set and Copernicus-Sentinel program data set. And choose to use Sentinel images because they have better resolution and sampling frequency. Can the author specify or suggest restrictions and conditions for satellite imagery?
- The authors have made some descriptions on the “Mine and dam discovery model” criteria, which is the focus of this paper. However, I personally think that, by transforming these texts into diagrams or tables, the authors can better present their ideas in a more logical and analytical way.
- The authors have provided many Deep Learning methods related information. For the purpose of result comparison, it is suggested that the authors can summarize these numerical data or brief information in a table.
- The authors’ purpose a new technologies to characterize illegal mining and the proliferation of tailings dams and to perhaps offer a powerful monitoring solution. The conclusions derived from the dataset results need to be clearer and be more connected to the subject in order to point out the importance of this study. In regards to data analysis, the authors need to be more detailed in analysis methods as they mainly presented the dataset results.
- Not many English mistakes were found in the paper. However, there are some minor mistakes that the authors will need to look at closely. Also, it is suggested that the authors can use semicolon to separate items in one sentence. This will help to make the paper even more readable.
Apart from the said concern, I would personally suggest that it is therefore recommended for acceptance on condition of minor revisions.
Author Response
- Some figures are not clear (e.g. Figure 1). It’s better to indicate the location of study area. And shows the orientation of a picture with letters (like E-W) inside the picture than with N arrows. They are much too early in the text, sometimes several pages before the mention in the text.
R: All satellite imagery has been improved, including orientation, scale and coordinates.
- The author mentioned two data sets, namely Landsat Mission data set and Copernicus-Sentinel program data set. And choose to use Sentinel images because they have better resolution and sampling frequency. Can the author specify or suggest restrictions and conditions for satellite imagery?
R: A paragraph was included in section 3.1 explaining spatial resolution and revisit time used to choose Sentinel-2 images for the project.
- The authors have made some descriptions on the “Mine and dam discovery model” criteria, which is the focus of this paper. However, I personally think that, by transforming these texts into diagrams or tables, the authors can better present their ideas in a more logical and analytical way.
R: Tables 2 and 3 have been improved and an explanation of the network architecture based on a sequence of layers has been included in the text.
- The authors have provided many Deep Learning methods related information. For the purpose of result comparison, it is suggested that the authors can summarize these numerical data or brief information in a table.
R: We added new accuracy metrics to sections 4.1 and 4.2 where the validation of the methods is presented: F1 score, precision, recall, ROC AUC and Kappa. These results and the corresponding confusion matrices are now presented in table form.
- The authors’ purpose a new technologies to characterize illegal mining and the proliferation of tailings dams and to perhaps offer a powerful monitoring solution. The conclusions derived from the dataset results need to be clearer and be more connected to the subject in order to point out the importance of this study. In regards to data analysis, the authors need to be more detailed in analysis methods as they mainly presented the dataset results.
R: A paragraph was added to the conclusion section summarizing the results obtained.
- Not many English mistakes were found in the paper. However, there are some minor mistakes that the authors will need to look at closely. Also, it is suggested that the authors can use semicolon to separate items in one sentence. This will help to make the paper even more readable.
R: A complete revision of the text was made.

Reviewer 2 Report
The authors of the manuscript entitled “Mining and Tailings Dam Detection In Satellite Imagery Using Deep Learning” present a novel approach to detect mining areas using deep learning and Sentinel-2 images.
The text is very well-written, and it is apparent that a lot of effort has been put into it. However, there are some issues that should be addressed:
- The aim and specific objectives of this study could be clearer. I recommend editing the lines 56 – 59 so that the aim and the specific objectives of the study are clearly defined.
- The section titles could follow a more conventional structure (i.e. Introduction, Background, Materials and methods, Discussion, Conclusion). This could make the navigation through the manuscript easier for the reader.
- The maps presented in the figures lack scale bar, north arrow, legend, and reference map (or coordinates on the edges of the map). Also, the captions of the figures could be more detailed, and include additional information about the figure. Please make the appropriate changes.
- The confusion matrices could be improved. Firstly, since confusion matrices are represented with tables, there should be an appropriate caption for each one. Also, it would be useful to calculate some more metrics (i.e. Kappa, user’s accuracy, producer’s accuracy), as well as the F1 score of the validation of the environmental impact classification model.
- In Figure 9, some of the yellow markers indicate the existence of a mine in places where seemingly no mine is present. Is this because of misclassification, or is there another reason?
- It is mentioned in the text that a significant number of false positives were present (L. 462 – 465), and that the red circles in Figure 11 indicate confirmed predictions while all the other dark areas are false positives. However, the confusion matrix (L. 120 – 121) for the validation of the mine and dam localization model shows that very few false positives are present. Please elaborate.
- Finaly, minor proofreading for grammatical errors (i.e. L. 120 – 121 [where does “he” refer to?], L. 130 – 132 [provides what?]) would improve the quality of the manuscript.
Author Response
The aim and specific objectives of this study could be clearer. I recommend editing the lines 56 – 59 so that the aim and the specific objectives of the study are clearly defined.
R: The last paragraph of section 1 has been reformulated to make the objectives of the work explicit.
· The section titles could follow a more conventional structure (i.e. Introduction, Background, Materials and methods, Discussion, Conclusion). This could make the navigation through the manuscript easier for the reader.
R: Section titles have been modified as suggested.
· The maps presented in the figures lack scale bar, north arrow, legend, and reference map (or coordinates on the edges of the map). Also, the captions of the figures could be more detailed, and include additional information about the figure. Please make the appropriate changes.
R: all maps were redone as suggested
· The confusion matrices could be improved. Firstly, since confusion matrices are represented with tables, there should be an appropriate caption for each one. Also, it would be useful to calculate some more metrics (i.e. Kappa, user’s accuracy, producer’s accuracy), as well as the F1 score of the validation of the environmental impact classification model.
R: The confusion matrices are now presented as tables and new metrics were added.
· In Figure 9, some of the yellow markers indicate the existence of a mine in places where seemingly no mine is present. Is this because of misclassification, or is there another reason?
R: This explanation was added to the text: “It is necessary to understand that each yellow pin points to the center of an implicit area where a mine or dam pattern was found. In this way, some marks are found in the vicinity of a mine or dam and not exactly in it because the pattern found in that specific prediction could be outside the center of the considered area. This does not mean that they are false positives. In this way, each mine or dam will be identified by a cloud of nearby marks indicating the region where the mine is located.”
· It is mentioned in the text that a significant number of false positives were present (L. 462 – 465), and that the red circles in Figure 11 indicate confirmed predictions while all the other dark areas are false positives. However, the confusion matrix (L. 120 – 121) for the validation of the mine and dam localization model shows that very few false positives are present. Please elaborate.
R: The average confusion matrix correspond to the tests made on the initial dataset (10 fold training and validation), acquired as explained in section 3.1 (using the coordinates of official mines and dams). Figure 11 is the result of the predictions made on a new large area followed by visual inspection where a number of non-official mines were found but also some false positives occurred. Section 4.3 and Figure 11 were modified to improve this explanation.
· Finaly, minor proofreading for grammatical errors (i.e. L. 120 – 121 [where does “he” refer to?], L. 130 – 132 [provides what?]) would improve the quality of the manuscript.
R: Done.

Round 2
Reviewer 2 Report
The authors of the manuscript entitled “Mining and Tailings Dam Detection In Satellite Imagery Using Deep Learning” have made most of the changes proposed by Reviewer 1. The manuscript is generally of high quality; however, some minor issues remain:
- North arrows, scale bars, and coordinate fonts differ across figures. I suggest using consistent notation across all maps. If necessary, such notation could be placed outside the maps.
- I would also suggest adding legends to the maps so that they are easier to read.
- Coordinates could be placed at the edges of the maps in the form of a grid, so specific coordinates correspond to specific points, thus making the maps more reader friendly.
- Figure 10 lacks scale bar, north arrow, coordinates, and a legend. Please add the appropriate notation.
- The north arrow is not visible in Figures 11 – 21. I propose using a different color or placing such notation outside the map.
Author Response
Most of the figures have been modified according to your suggestions.
Figure 9 was kept unchanged as its objective is to show that the results of the predictions can be inspected in Google Earth, whose interface for presenting maps is different from the interface of the GIS environment and does not allow many customizations.
